# Sea Urchin-like Si@MnO_2_@rGO as Anodes for High-Performance Lithium-Ion Batteries

**DOI:** 10.3390/nano12020285

**Published:** 2022-01-17

**Authors:** Jiajun Liu, Meng Wang, Qi Wang, Xishan Zhao, Yutong Song, Tianming Zhao, Jing Sun

**Affiliations:** College of Sciences, Northeastern University, Shenyang 110819, China; 2000165@stu.neu.edu.cn (J.L.); 2000176@stu.neu.edu.cn (M.W.); wangqi@mail.neu.edu.cn (Q.W.); 2000189@stu.neu.edu.cn (X.Z.); 2000172@stu.neu.edu.cn (Y.S.); zhaotm@stumail.neu.edu.cn (T.Z.)

**Keywords:** Si, MnO_2_, rGO, sea urchin-like structure, lithium-ion battery, high performance

## Abstract

Si is a promising material for applications as a high-capacity anode material of lithium-ion batteries. However, volume expansion, poor electrical conductivity, and a short cycle life during the charging/discharging process limit the commercial use. In this paper, new ternary composites of sea urchin-like Si@MnO_2_@reduced graphene oxide (rGO) prepared by a simple, low-cost chemical method are presented. These can effectively reduce the volume change of Si, extend the cycle life, and increase the lithium-ion battery capacity due to the dual protection of MnO_2_ and rGO. The sea urchin-like Si@MnO_2_@rGO anode shows a discharge specific capacity of 1282.72 mAh g^−1^ under a test current of 1 A g^−1^ after 1000 cycles and excellent chemical performance at different current densities. Moreover, the volume expansion of sea urchin-like Si@MnO_2_@rGO anode material is ~50% after 150 cycles, which is much less than the volume expansion of Si (300%). This anode material is economical and environmentally friendly and this work made efforts to develop efficient methods to store clean energy and achieve carbon neutrality.

## 1. Introduction

In recent years, environmental pollution caused by carbon emissions has an increasing urgency for developing high-density, long lifetime storage materials or storage devices for clean energy. As a carrier of clean energy, lithium-ion batteries play a pivotal role in the country’s goal of achieving carbon neutrality [1,2,3,4]. A Silicon-based electrode is one of the most promising candidates as an anode for lithium-ion batteries and is expected to replace the use of a commercial graphite electrode (372 mAh g^−1^) due to its remarkable theoretical capacity (4200 mAh g^−1^) [5,6]. It is also widely regarded for its good voltage platform, environmental friendliness, and abundant reserves. Despite these advantages, Si-based lithium-ion batteries still face severe volume expansion during the charging/discharging process, poor electrical conductivity, and a short cycle life [7,8]. The safety hazards and unstable performances of Si materials resist the applications of lithium-ion batteries in commercial use. Therefore, many attempts on modifying Si materials have been made to restrict the volume expansion and enhance electrical conductivity for improving the performance of Si-based lithium-ion batteries [9,10,11,12].

The rational design of an anode has been considered as an effective strategy to enhance Si-based lithium-ion batteries’ performances [13,14]. Some studies have reported that coating or doping can effectively reduce the large volume changes and the subsequent accumulation of excessive stress during lithiation–delithiation cycles [15,16,17,18,19]. Especially, Si nanospheres are combined with carbon materials, a strategy that aims to provide extra space to accommodate volumes’ expansion and improve the electrical conductivity of the electrode, such as Si/ reduced graphene oxide (rGO) and Si/Carbon Nanotube (CNT) [20,21,22]. However, the size gap between Si nanospheres and rGO sheets is large, and Si nanospheres tend to agglomerate and do not easily migrate uniformly into the rGO sheets to form a stable structure. It is worth mentioning that transition metal oxides are also good anodes with high theoretical capacity, such as MnO_2_ (1233 mAh g^−1^) [23,24,25,26,27,28]. In addition, the volume changes of MnO_2_ in lithium-ion intercalation and deintercalation is very small. A reasonable coating structure was used to grow MnO_2_ evenly outside the Si nanospheres, which can avoid the material crushing and form an unstable solid electrolyte interface (SEI) film due to the volume expansion of Si nanospheres during the circulation process, lowering the capacity. At the same time, rGO has high electrical conductivity and excellent physical properties. The low conductivity of transition metal oxides and Si nanospheres can be compensated by rGO. MnO_2_ wraps the Si nanosphere to avoid the agglomeration of the Si nanosphere, increase the contact area with the rGO sheet, ensure the interface strength between MnO_2_ and rGO, and improve the structural stability of the material [29,30,31,32].

Herein, we propose a simple hydrothermal method to produce sea urchin-like Si@MnO_2_@rGO as anodes. In this unique structure, MnO_2_ and rGO surrounding Si nanospheres formed a strong armor, relieving the mechanical strain generated by the volume expansion of the Si nanospheres and providing enough space to buffer, which helps provide to a long lifetime. In addition, the sea urchin-like Si@MnO_2_@rGO anode showed an initial discharge capacity of 1378.15 mAh g^−1^ at a current density of 0.1 A g^−1^ and a discharge specific capacity was maintained at 1282.72 mAh g^−1^ under a test current of 1 A g^−1^ after more than 1000 cycles, showing excellent chemical performance at different current densities. Therefore, this rational design provides a new route for the development of high-performance Si-based anodes; this work made efforts to store clean energy and achieve carbon neutrality [33,34].

## 2. Materials and Methods

### 2.1. Materials

All chemicals were analytical reagent grade and used as received. Graphene oxide (GO) was purchased from XFNANO Materials Tech Co., Ltd, Nanjing, China. Si nanospheres, ethanol, MnSO_4_, KMnO_4_, KH550, and NaBH_4_ were obtained from National Medicines Corporation Ltd, Shanghai, China.

### 2.2. Preparation of Si@MnO_2_@rGO

First, 0.5 g of GO and 0.1 of M NaBH_4_ were ultrasonically dispersed in 100 mL of deionized water with stirring treatment at 80 °C for 24 h. After cooling to room temperature, the sample was centrifuged to obtain reduced graphene oxide (rGO). The above-prepared rGO of 0.05 g was dispersed in 20 mL of deionized water and sonicated for 30 min, denoted as Solution A.

Then, 0.086 g of Si nanospheres (50 nm) were dispersed in a beaker containing 20 mL of water and 20 mL of alcohol, followed by the addition of 50 μL of KH550 silane coupling reagent. The sample was sonicated for 30 min to form a homogeneous solution, denoted as Solution B. Next, Solution A, Solution B, 0.64 g of MnSO_4,_ and 1 g of KMnO_4_ were sequentially added in a reaction vessel and stirred with magnetic force for 2 h. The reaction temperatures were set as 20, 50, or 160 °C, respectively. After cooling to room temperature, the products were centrifuged with water and ethanol several times to remove the residual reaction products. A freeze-dried treatment was used to remove the water. Finally, the samples were denoted as Si@MnO_2_@rGO-20°C, Si@MnO_2_@rGO-50°C, and Si@MnO_2_@rGO-160°C, respectively.

The synthesis of Si@MnO_2_ was similar to the synthesis Si@MnO_2_@rGO, only missing the rGO, denoted as Si@MnO_2_-20°C, Si@MnO_2_-50°C, and Si@MnO_2_-160°C, respectively.

### 2.3. Electrochemical Measurements

The homogeneous slurry was prepared by mixing Si@MnO_2_@rGO, acetylene black, and polyvinylidene fluoride (PVDF) in N-methyl-2-pyrrolidone (NMP) with a mass ratio of 8:1:1. After fully stirring, the prepared slurry was evenly coated on a copper foil by a sputter coater, whose space was 25 μm. The dimeter of electrode was 1.5 cm and the anode material mass load was ~1 mg. The electrolyte of each coin cell was ~45 μL. Then, the electrode sheet was placed into a vacuum drying oven for 12 h at 80 °C. The working electrode was the prepared electrode sheet (Si@MnO_2_@rGO composites). The electrolyte was a LiPF_6_ (1.0 mol L^−1^) in a 1:1:1 v/v/v mixture of ethylene carbonate, dimethyl carbonate, and ethyl methyl carbonate. The separator was Celgard 2400 (Saibo, Beijing, China). The capacity calculation and cycling rate were set by a battery testing system (CT3008, Kejing, Hefei, China). The galvanostatic charge/discharge (GCD) tests were conducted in the voltage window of 0.1–3.2 V. The cyclic voltammetry curve (CV) was carried out with scan rate of 0.1 mV s^−1^ between the voltage range of 0.1–3.2 V using an electrochemical workstation (CHI 600E, Chenhua, Shanghai, China). The electrochemical impedance spectroscopy (EIS) was performed over the frequency range from 100 kHz to 0.1 Hz with an alternating current (AC) impedance of 5 mV and was also recorded by an electrochemical workstation (CHI 600E, Chenhua, Shanghai, China).

### 2.4. Materials’ Characterization

Field-emission scanning electron microscopy (FE-SEM; SU8010, Hitachi High-Tech, Tokyo, Japan) and field-emission transmission electron microscopy (FE-TEM; JEM2100F, JEOL, Tokyo, Japan) were used to characterize the morphology and elemental distribution of the electrode materials. X-ray diffraction (XRD; Rigaku lnc., Tokyo, Japan) was used to characterize the phases, crystallinity, and crystal structures of the samples. The species and chemical composition of the surface elements of the samples were analyzed using X-ray photoelectron spectroscopy (XPS, Thermo AXIS-SUPRA, Kratos, Manchester, UK).

## 3. Results and Discussion

### 3.1. Anode Material Design and Morphology Characterization

Figure 1 shows a schematic of the preparation of the sea urchin-like Si@MnO_2_@rGO composites. First, GO sheets were reduced to rGO by a hydrothermal method with NaBH_4_. Then, the rGO was collected and dispersed in deionized water (Solution A). Next, Si nanospheres were dispersed in a solution of water, alcohol, and the KH550 silane coupling agent (Solution B). The composites were well grafted by KH550, improving the conductivity. Finally, Solution A, Solution B, KMnO_4,_ and MnSO_4_ were added in a reaction vessel and stirred with magnetic force for 2 h. The reactions were carried out at different temperatures, resulting in different morphologies. Si nanospheres were coated with MnO_2_ and rGO, which may have contributed to the KH550. The unique double-layer structure of MnO_2_ and rGO effectively mitigated the volume expansion of Si nanospheres during the charging/discharging process. In addition, rGO had excellent electrical conductivity, compensating for the low electrical conductivity of Si nanospheres and MnO_2_. This was the basis of the rational design of the Si-based anode material for enhancing lithium-ion battery performance.

Figure 2 shows the morphologies of the materials grown at three different temperatures. The morphology of Si@MnO_2_-20°C is shown in Figure 2a. The MnO_2_ formed disordered and entangled filaments on the surface of Si nanospheres. Figure 2b shows Si@MnO_2_-20°C at a high magnification. The surface of the material was rough and the diameter of Si@MnO_2_-20°C particles was 200 nm. Figure 2c is an SEM image of Si@MnO_2_@rGO-20°C, showing that the rGO covered the Si@MnO_2_-20°C. However, some MnO_2_ nanowires appeared on the surface of rGO, which suggested that the material was not stable. Figure 2d,e show the SEM images of Si@MnO_2_-50°C at different magnifications, respectively. As shown in Figure 2d, the MnO_2_ formed a sea urchin-like shell coating on the surface of Si nanospheres. Figure 2e shows the sea urchin-like Si@MnO_2_-50°C. The stings (MnO_2_) were 100–300 nm in length. Figure 2f shows Si@MnO_2_@rGO-50°C. The rGO wrapped the Si@MnO_2_-50°C without obvious damage to the structure. Interestingly, this rGO surface was clean, without any MnO_2_ nanowires, indicating the sea urchin-like structure was more stable. MnO_2_ nanowires were vertically oriented on the outer layer of the Si nanospheres, imbedding rGO layers, forming a stable structure, which can buffer the excessive expansion of the Si nanospheres during the charging/discharging process and extend the batteries’ lifetime. Figure 2g shows the SEM image of Si@MnO_2_-160°C. The MnO_2_ nanowires grew longer and coated the surface of the Si nanospheres. Figure 2h is the enlarged view of Figure 2g. The plentiful MnO_2_ nanowires were interleaved. Figure 2i shows the images of Si@MnO_2_@rGO-160°C. It can be clearly seen that the MnO_2_ nanowires were broken and dispersed on the surface of rGO, which indicates that the Si@MnO_2_@rGO-160°C was not stable enough. The stability of the material is one of the important factors of the lithium-ion batteries’ performance. The high stability of the anode material may suggest the long life-time. Therefore, the Si@MnO_2_@rGO-50°C was chosen for the following test.

Figure 3a shows the Energy Dispersive Spectrometer (EDS) analysis of the ball-milled material, using aluminum foil as a substrate, and the inset shows the atomic percentages of elements in the materials. The inset shows that the Si@MnO_2_@rGO-50°C contained ~48% Si, ~21% rGO, and ~31% MnO_2_. Appendix A shows the capacity contribution percentages of anode materials (Si, ~63.3%; MnO_2_, ~35.6%; and rGO, ~1.1%). The existence of MnO_2_ can not only resist the volume expansion of Si, but also provide the capacity contribution. Figure 3b,c show HRTEM images of the Si@MnO_2_@rGO-50°C composite at different magnifications. Si nanospheres were coated with rGO and MnO_2_. The lattice fringes corresponded to the (111) plane of Si, having a separation of 0.31 nm [35,36]. In addition, lattice fringes having an interplanar spacing of 0.47 nm can be seen, and this is consistent with the (200) plane of MnO_2_ [25,37,38]. The element mapping of the Si@MnO_2_@rGO-50°C composite is shown in Figure 3d–h. Mn element and O element contributed to MnO_2;_ Si element contributed to Si nanospheres. C element dispersed throughout the image may have contributed to both the substrate and rGO.

Figure 4a shows the XRD patterns of rGO, Si, Si@MnO_2_-50°C, and Si@MnO_2_@rGO-50°C, respectively. In the XRD pattern of Si@MnO_2_-50°C, the pronounced peaks at 28.5°, 47.4°, 56.2°,58.9°, 69.2°, 76.5°, and 88.2° corresponded to the (111), (220), (311), (222), (400), (331), and (422) planes of Si (PDF#77-2108). In addition, the characteristic peaks at 12.7°, 18.0°, 28.7°, 37.6°, 41.1°, 49.8°, 59.5°, 65.5°, 68.5°, and 72.5° corresponded to the (110), (200), (310), (121), (420), (411), (260), (002), (202), and (631) planes of MnO_2_ (PDF#72-1982). The XRD pattern of the Si@MnO_2_@rGO-50°C composites contained peaks corresponding to Si and MnO_2_, as well as the broad peak ranges from 20° to 30°, which were indexed to the standard peaks of rGO [39,40,41,42]. Figure 4b shows the XPS spectrum of the Si@MnO_2_@rGO-50°C composites, which revealed the presence of Si, Mn, O, and C, corresponding to element mapping (Figure 3d–g). Figure 4c,d shows the high-resolution XPS spectra. Figure 4c shows the Si 2p spectrum, whose peak at 99.7 eV was related to Si-Si bonds; two small peaks at 101.9 and 103.6 eV contributed to organic Si and Si-O, respectively, which may have been caused by the slight oxidation of Si in the thermal-treated process. Figure 4d shows the Mn 2p spectrum, which contained two spin–orbit peaks corresponding to Mn 2p_3/2_ (642.5 eV) and Mn 2p_1/2_ (654.1 eV) of MnO_2_, whose separation between these peaks was 11.6 eV.

### 3.2. Lithium-Ion Battery Performance

A CV experiment carried out to further evaluate the lithium storage behavior is shown in Figure 5. Figure 5a shows the CV curves of Si@MnO_2_@rGO-50°C as the independent anodes of lithium-ion batteries for the first four cycles at a scan rate of 0.1 mV s^−1^ between 0.1 V and 3.2 V. In the first cycle, a clear cathodic peak at 0.16 V corresponded to the lithium alloying process of crystalline Si and the formation of an amorphous Li*_x_*Si phase; a clear cathodic peak at 0.10 V corresponded to the formation of the Li_15_Si_4_ phase. An anodic peak at 0.24 V was related to the delithiation of Li_15_Si_4_; an anodic peak at 0.50 V was related to the transition from the Li*_x_*Si phase to amorphous Si, according to Equations (1) and (2).
(1)$$x{\mathrm{Li}}^{+}+\mathrm{Si}+xe^{-}↔{\mathrm{Li}}_{x}\mathrm{Si},$$
(2)$${\mathrm{Li}}_{x}\mathrm{Si}+x{\mathrm{Li}}^{+}+xe^{-}↔{\mathrm{Li}}_{15}{\mathrm{Si}}_{4},$$

The redox peaks of Si coincided with those reported previously [43,44,45]. In addition, the reversible redox peaks at 1.23 and 0.36 V, respectively, were consistent with the lithiation and delithiation reactions of MnO_2,_ according to Equation (3). When the cathodic peak was 0.36 V, Li was inserted into the anode to form LiO_2_ and MnO_2_ was reduced to Mn. An anodic peak at 1.23 V was related to the charging process of the lithium ion battery; Mn can facilitate the decomposition of LiO_2_.
(3)$${\mathrm{MnO}}_{2}+4\;\mathrm{Li}↔2{\;\mathrm{Li}}_{2}O+\mathrm{Mn},$$

In the subsequent scans, the redox peaks were largely coincident, which was attributed to the stability of the Si@MnO_2_@rGO-50°C, indicating the good electrochemical reversibility for lithium-ion batteries.

Figure 5b shows the galvanostatic discharge/charge curves of Si@MnO_2_@rGO-50°C at 0.1 A g^−1^ over the range between 0.1 V and 3.2 V. In the first cycle, the initial discharge capacity was 1378.15 mAh g^−1^. In the second, third, fifth, and 10th cycles, the discharge specific capacities were 1279.21, 1208.02, 1150.74, and 1093.70 mAh g^−1^, respectively. In the subsequent cycles, the charge and discharge curves basically overlapped, which indicated good capacity retention. In addition, the plateau of Si at 0.5 V in the figure was consistent with the CV results.

Figure 5c shows the cycling performances of the anode (Si, Si@MnO_2_-50°C, Si@MnO_2_@rGO-20°C, Si@MnO_2_@rGO-50°C, and Si@MnO_2_@rGO-160°C) at a current density of 0.1 A g^−1^. Of these samples, Si@MnO_2_@rGO-50°C showed the best cycling performance. The initial discharge specific capacity of Si was 1855.62 mAh g^−1^. However, after 20 cycles, the discharge specific capacity decayed to 330.03 mAh g^−1^; after 150 cycles, the discharge specific capacity was almost 0. This is because, during the cycling process, the slurry of the active material became dislodged from the collector due to the serious volume expansion of Si nanospheres caused by lithium-ion intercalation and deintercalation. When the Si nanospheres underwent volume changes during the charging/discharging process, the formed SEI film was broken, resulting in new surfaces being exposed in the electrolyte. The exposed surfaces needed external lithium ions to form a stable SEI film, which led to a dramatic decrease in capacity. Although Si@MnO_2_-50°C had a higher discharge specific capacity than Si nanospheres alone after 150 cycles, it still did not meet current demands for battery energy storage. The initial specific capacity of Si@rGO-50°C was 1180.41 mAh g^−1^; after 150 cycles, the capacity was maintained at 543.84 mAh g^−1^ (Figure 5c). Compared with Si@rGO-50°C and Si@MnO_2_@ rGO-50°C, the existence of MnO_2_ improved the specific capacity, which may have been due to the synergistic effect of MnO_2_ and rGO. The initial specific capacities of Si@MnO_2_@rGO-20°C and Si@MnO_2_@rGO-160°C were 1670.24 and 2450.32 mAh g^−1^, respectively. Furthermore, after 100 cycles, the specific capacities were 512.14 and 665.19 mAh g^−1^, respectively, with low capacity retention rates. The initial specific capacity of Si@MnO_2_@rGO-50°C was 1378.14 mAh g^−1^; after 150 cycles, the capacity was maintained at 960.21 mAh g^−1^. Among Si@MnO_2_@rGO-50°C, Si@MnO_2_@rGO-20°C, and Si@MnO_2_@rGO-160°C, although the Si@MnO_2_@rGO-50°C exhibited a lower capacity, it had the excellent cyclability, which can better meet the commercial need of long lifetime, due to the stability of the sea urchin-like Si@MnO_2_@rGO-50°C and the dual protection of rGO and MnO_2_. Interestingly, the structure of rGO encapsulating the sea urchin-like Si@MnO_2_-50°C mitigated the volume-change effects of Si nanospheres, improving the electrical conductivity and contributing to the high capacity when comparing the Si and Si@MnO_2_-50°C. A comparison of the rate and cycling performances of the sea urchin-like Si@MnO_2_@rGO-50°C, at different various current densities, is shown in Figure 5d. At current densities of 0.1, 0.2, 0.5, 1, and 2 A g^−1^, the specific charging capacities were 1323.87, 971.85, 701.12, 491.85, and 272.47 mAh g^−1^, respectively. Furthermore, when the current density rose again to 0.1 A g^−1^, the specific charging capacity recovered to 981.68 mAh g^−^^1^ and the capacity retention rate was 74.2%. Compared to previous reports of Si and Si@rGO, Si@MnO_2_@rGO-50°C demonstrated a long lifetime and high capacity retention rates [46,47]. Furthermore, to evaluate the cycling stability at high current densities, Si@MnO_2_@rGO-50°C was tested at 1 Ah g^−1^ (Figure 5e). In the first charge/discharge cycle, the specific capacity was 1446.85 mAh g^−1^; after 1000 cycles, the specific capacity was 1282.72 mAh g^−1^ with a coulombic efficiency of 99.4%. The composite material also showed excellent cycling performance at high currents. The decrease in capacity fluctuations at the first 80 cycles was due to the irreversible formation of SEI film on the materials’ surface. Interestingly, the capacity exhibited an increasing trend from the ~80th cycle upwards. This was attributed to the reversible growth of a polymeric gel-like film. After a long-cycling charging/discharging process, the polymeric gel-like film gradually degraded. The improved electrode kinetics increased the capacity. This phenomenon has been widely reported among transition metal oxides [48,49,50,51,52,53,54,55]. Appendix A shows the performance comparison among the reported works and indicates the high capacity and long lifetime of Si@MnO_2_@rGO-50°C. When the cycle reached 150 times, the specific capacity was 960.21 mAh g^−1^ at 0.1 A g^−1^ and 746.13 mA g^−1^ at 1 A g^−1^. This is because the higher the charge/discharge current density was, the fast the electrode chemical reaction speed became. A large number of lithium ions reacted on the surface of the anode materials instantly, leading to the formation of concentration polarization on the electrode surface. Part of the active materials had no time to react, and the utilization rate of the active materials became smaller, resulting in the decreasing in capacity.

EIS measurements were carried out to further investigate the electrochemical mechanism shown in Figure 6. Figure 6a shows the EIS plots and corresponding fitting plots of Si, Si@MnO_2_-50°C, and Si@MnO_2_@rGO-50°C, respectively. Additionally, the inset shows the data equivalent circuit diagrams. The EIS diagram consists of a semicircle in the mid-high-frequency region and a diagonal line in the low-frequency region. Here, R_CT_ in the mid-frequency region corresponds to the charge-transfer impedance and W_O_ in the low-frequency region corresponds to the diffusion of Li^+^ inside the electrode material [56,57]. The R_S_ and R_CT_ values fitted from the equivalent circuit model are summarized in Appendix A for comparison. The Si@MnO_2_@rGO-50°C exhibited a low value of R_CT_ (146.2 Ω) before cycling, which was the lowest among Si, Si@MnO_2_@rGO-50°C, and Si@MnO_2_-50°C. This result was attributed to the rGO networks, prominently improving the electronic conductivity of the electrodes and lowering the charge transfer impedance. Figure 6b shows the EIS plots and corresponding fitting plots of Si@MnO_2_@rGO-50°C before and after 150 cycles at 0.1 A g^−1^; cycled Si@MnO_2_@rGO-50°C was fitted with the same equivalent circuit model in Figure 6a. The R_CT_ (72.7 Ω) was lower after cycling, which implies the charge transfer impedance decreased substantially, demonstrating the stable SEI film and good structure stability during cycling of Si@MnO_2_@rGO-50°C.

Figure 7a,b shows the cross-sectional SEM images of Si before and after 150 cycles at 0.1 A g^−1^. After cycles, the volume expansion was ~323% (from 5.3 μm to 22.41 μm). The huge change in volume expansion of Si may have contributed to the intercalation and deintercalation of lithium ions during cycles and the exfoliation of the active materials. As shown in Appendix A, the volume expansion of Si@MnO_2_-50°C electrode after 150 cycles at 0.1 A g^−1^ was ~198% (from 4.29 μm to 12.8 μm). Interestingly, the volume expansion of Si@MnO_2_@rGO-50°C electrode was ~59% (8.13 μm to 12.9 μm) after 150 cycles at 0.1 A g^−1^. Such a small volume change in the active materials guarantees long-term cycling stability. Appendix A shows the top-view SEM image of Si@MnO_2_@rGO-50°C after 150 cycles at 0.1 A g^−1^. The electrode sheet was relatively intact without signs of rupture. Appendix A shows the element mapping of Appendix A, suggesting Si, Mn, O, and C were evenly distributed. The mechanism of the Si electrode reaction was further illustrated in Figure 7e. The lithiation of Si resulted in the formation of an amorphous silicon–lithium alloy (Li*_x_*Si) (0.16 V). Then, the amorphous Li*_x_*Si was transformed to crystalline Li_15_Si_4_ (0.1 V). The delithiation process involved the transformation from crystalline Li_15_Si_4_ to amorphous Li*_x_*Si (0.24 V) and, finally, to amorphous Si (0.5 V) [58,59,60].

## 4. Conclusions

In summary, sea urchin-like Si@MnO_2_@rGO-50°C as an anode for lithium-ion batteries was presented. The reversible capacity, cyclability, and rate capability were very high, which was attributed to the dual protection of rGO and MnO_2_. The discharge specific capacity was maintained at 1282.72 mAh g^−1^ under a test current of 1 A g^−1^ after more than 1000 cycles. Such high cyclability may be attributed to the sea urchin-like structure reducing the volume expansion of anodes during the charging/discharging process. The present results indicate that the sea urchin-like Si@MnO_2_@rGO-50°C is a good candidate for high performance anodes of lithium-ion batteries. This work made efforts to develop efficient methods to store clean energy and achieve carbon neutrality.

## Figures and Tables

**Figure 1 nanomaterials-12-00285-f001:**
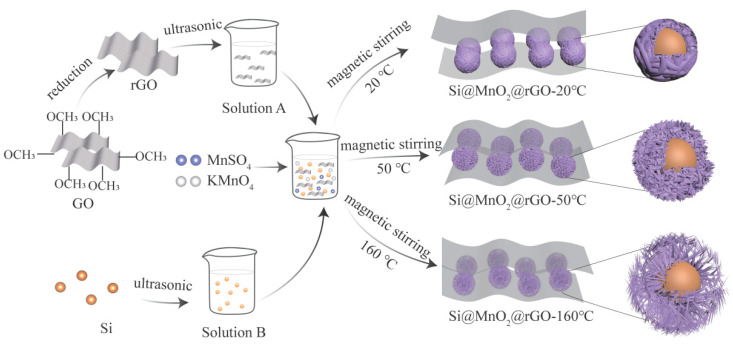
Concise preparation schematics of the Si@MnO_2_@ reduced graphene oxide (rGO) composites.

**Figure 2 nanomaterials-12-00285-f002:**
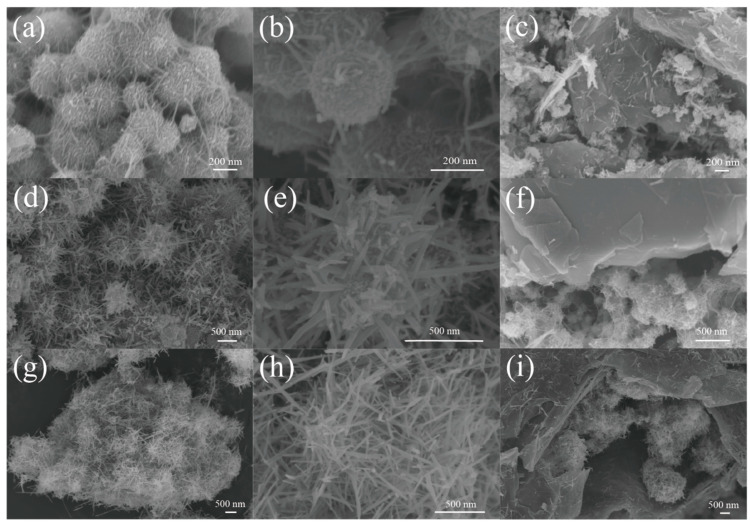
(**a**) SEM image of the Si@MnO_2_-20°C composite. (**b**) Enlarged view of the SEM image of the Si@MnO_2_-20°C composite. (**c**) SEM image of the Si@MnO_2_@rGO-20°C composite. (**d**) SEM image of the Si@MnO_2_-50°C composite. (**e**) Enlarged view of the SEM image of the Si@MnO_2_-50°C composite. (**f**) SEM image of the Si@MnO_2_@rGO-50°C composite. (**g**) SEM image of the Si@MnO_2_-160°C composite. (**h**) Enlarged view of the SEM image of the Si@MnO_2_-160°C composite. (**i**) SEM image of the Si@MnO_2_@rGO-160°C composite.

**Figure 3 nanomaterials-12-00285-f003:**
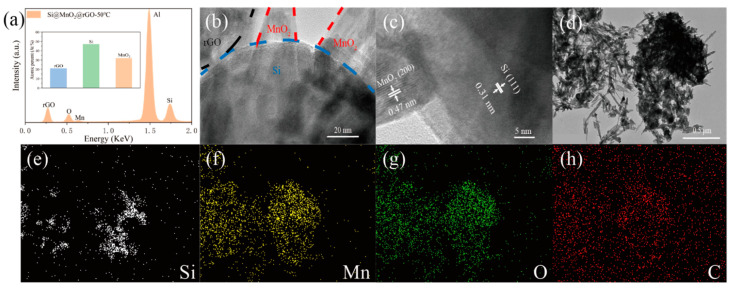
(**a**) Energy Dispersive Spectrometer (EDS) spectra of Si@MnO_2_@rGO-50°C. Inset showing the respective substance of Si@MnO_2_@rGO-50°C. (**b**,**c**) HRTEM images of the Si@MnO_2_@rGO-50°C composite at different magnifications. (**d**–**h**) Elemental mapping of the Si@MnO_2_@rGO-50°C composite.

**Figure 4 nanomaterials-12-00285-f004:**
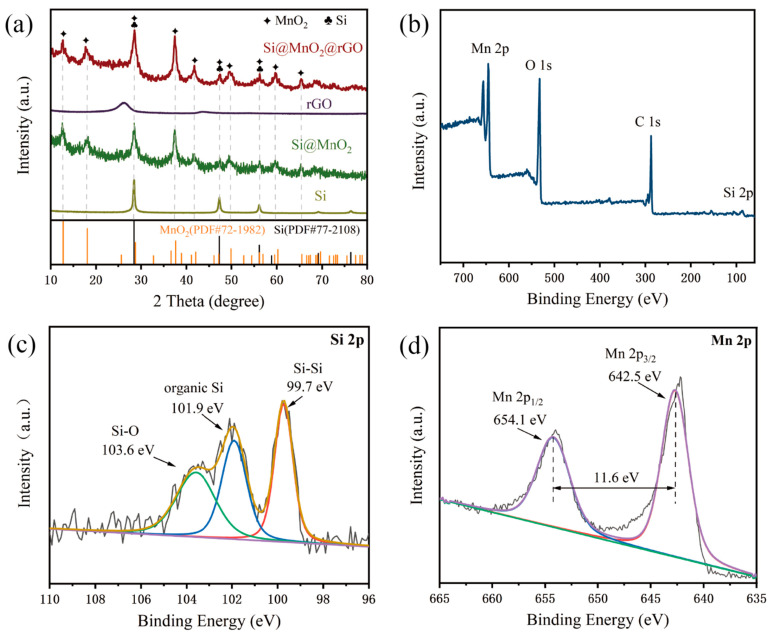
(**a**) X-ray diffraction (XRD) patterns of the Si, Si@MnO_2_-50°C rGO, and Si@MnO_2_@rGO-50°C, respectively. (**b**–**d**) XPS spectra of the Si@MnO_2_@rGO-50°C.

**Figure 5 nanomaterials-12-00285-f005:**
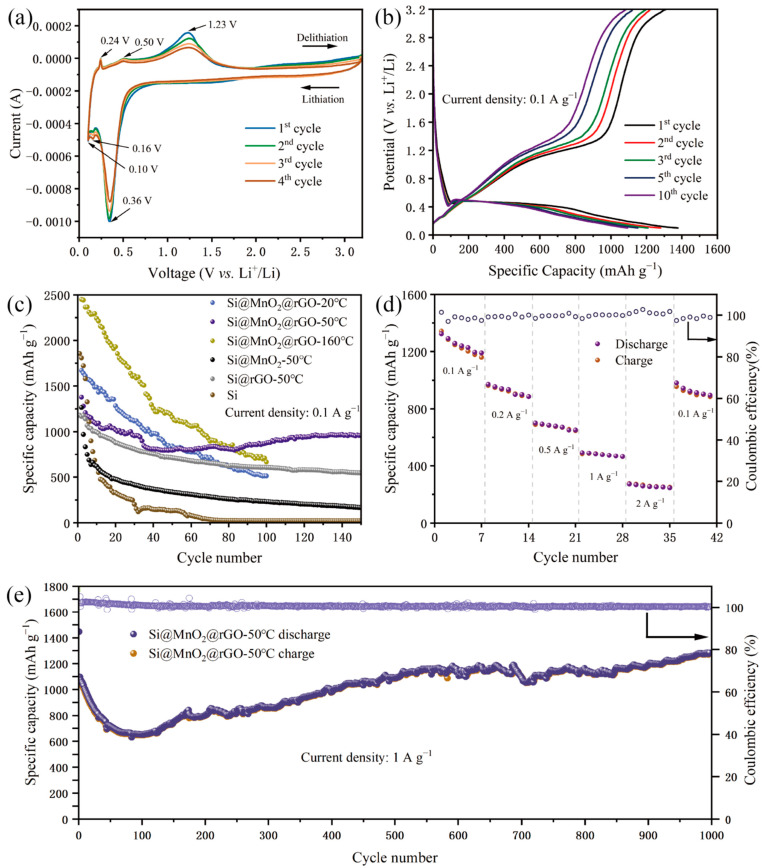
(**a**) CV curves of the Si@MnO_2_@rGO-50°C at 0.1 mV s^−1^ between 0.1 V and 3.2 V. (**b**) Galvanostatic charge–discharge curves of the Si@MnO_2_@rGO-50°C at 0.1 A g^−1^. (**c**) Long-term cycling performance of the Si, Si@MnO_2_-50°C, Si@rGO-50°C, Si@MnO_2_@rGO-20°C, Si@MnO_2_@rGO-50°C, and Si@MnO_2_@rGO-160°C at 0.1 A g^−1^. (**d**) Rate performance of the Si@MnO_2_@rGO-50°C. (**e**) Long-term cycling performance of the Si@MnO_2_@rGO-50°C at 1 A g^−1^.

**Figure 6 nanomaterials-12-00285-f006:**
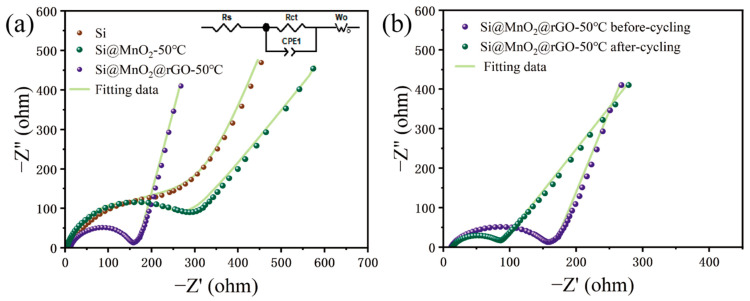
(**a**) The electrochemical impedance spectra of Si, Si@MnO_2_-50°C, and Si@MnO_2_@rGO-50°C. The inset shows the equivalent circuit model. (**b**) The electrochemical impedance spectra of the Si@MnO_2_@rGO-50°C before and after cycling.

**Figure 7 nanomaterials-12-00285-f007:**
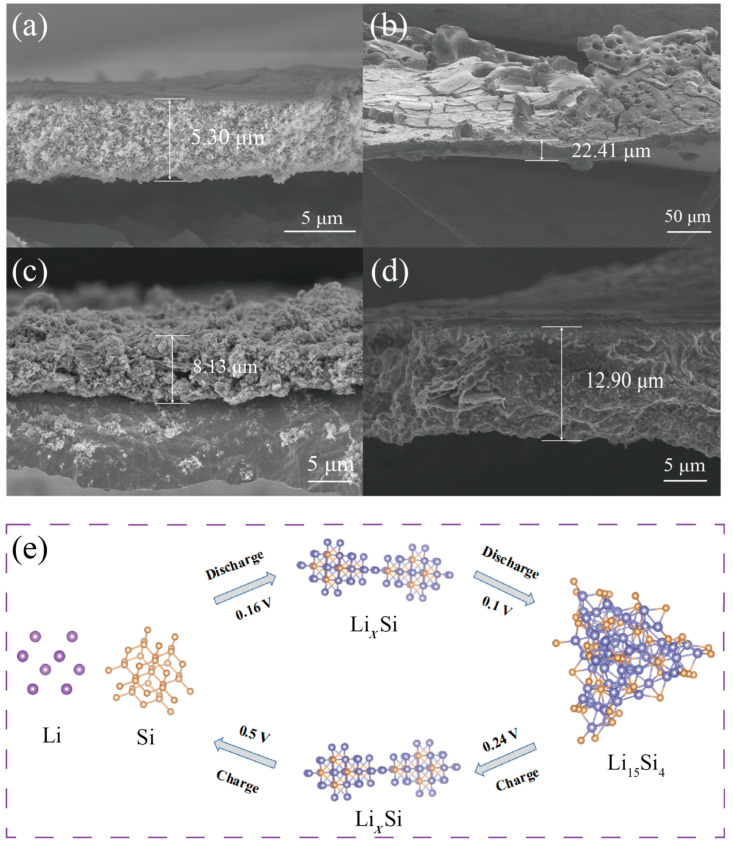
(**a**,**b**) Cross-sectional SEM images of Si before and after 150 cycles. (**c**,**d**) Cross-sectional SEM images of Si@MnO_2_@rGO-50°C before and after 150 cycles. (**e**) The mechanism of intercalation and deintercalation of lithium ions.

## Data Availability

The data presented in this study are available on request from the corresponding author.

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
