# Peer review of "Sea Urchin-like Si@MnO2@rGO as Anodes for High-Performance Lithium-Ion Batteries"

_nanomaterials, 2022, doi:10.3390/nano12020285_

Round 1
Reviewer 1 Report
Liu et al studied Si@MnO2@rGO with sea-urchin-like structure for the high-performance LIB anode. In this study, MnO2 and rGO function as dual protection layer for the Si active nanospheres that can undergo huge volume change during electrochemical reactions. More specifically, sea-urchin-like MnO2 rod shell (100-300 nm in length) vertically grown on Si nanosphere seems to provide the spacer between active Si nanosphere which prevents the agglomeration of Si nanoparticles and rGO provides conductive buffering matrix. Because of this synergistic effect, Si@MnO2@rGO showed good cyclic performance (1282 mAh g-1 at 1 A g-1 after 1000 cycles) and rate capability. The authors have carried out a systematic study on the new anode material by comparing the performance of various control samples and different heat treatment temperature conditions. However, before further decision, there are some issues to be addressed:
1) (Page 5) From EDS analysis, the individual contents of Si, MnO2, and rGO in Si@MnO2@rGO-50 are 48%, 31%, and 21%, respectively. Considering the high theoretical capacity of MnO2 (1233 mAh g-1), the capacity contribution of MnO2 cannot be negligible. However, the authors claimed that Si is the only major active material. Thus, I think authors should provide quantitative percentage of capacity contribution from Si, MnO2, and rGO and discuss whether the capacity contribution from MnO2 is really negligible.
2) Regarding the above issue, in order to understand the role of MnO2 more clearly, I suggest authors newly prepare another control sample without MnO2 (e.g., Si@rGO-50C) and compare the cyclic performance of this sample in Figure 5c.
3) (page 6) In Figure 4a, the title of x-axis (“2 Theta degree”) should be corrected.
4) (page 8) In Figure 5d, if possible, please compare the rate performance of Si@MnO2@rGO-50C with other control samples.
5) (page 10) In Figure 7, if the capacity contribution of MnO2 is not negligible, the electrochemical reaction mechanism of MnO2 needs to be added.
6) There are several grammatical errors. I suggest authors polish English.
Reviewer 2 Report
The submission of Article nanomaterials-1559350 titled, “Sea-urchin-like Si@MnO2@rGO as anodes for high-performance lithium-ion batteries” reports the modification of Si anode with a design of a ternary composite as Si@MnO2@rGO. The MnO2 and rGO are reported to protect the Si anode active material during cell cycling. The Si@MnO2@rGO has a capacity of 1282.72 mAh g-1 at a current density of 1 A g-1.
(1) In Materials and Methods, the thickness of the slurry is reported as 25 μm. Did the thickness of slurry analyze by facilities or mark by the blade? What is the thickness of anode material on the dried electrode? The size of the electrode was not reported. What is the mass loading of the anode material? What is the amount of electrolyte used in the coin cell?
(2) In Materials and Methods, the information for conducting the Electrochemical measurements should be reported here, such as (i) the calculation method of the capacity and the cycling rate (recommended, because of MnO2 and Si both involves in the electrochemical reactions), (ii) the analysis voltage window used for CV and GCD, (iii) and the frequency and AC used for EIS.
(3) In Results and Discussion, the reversible redox peaks at 1.23 and 0.36 V are reported that belong to the lithiation and delithiation reactions of MnO2. Thus, more discussion about the active material and the capacity are needed. It is suggested to clarify the calculation of the specific capacity per gram that is based on what in Materials and Methods. The capacity contributed by MnO2 should need additional discussion. A table summarizes this information is suggested.
(4) The CV curves of the Si@MnO2@rGO-50°C are reported that are analyzed at 0.1 mV s−1 between 0.1 V and 3.2 V. However, after the initial CV scanning, the voltage changes from 3.2 to 3.1V. Why?
(5) The EIS data are suggested to be compared with the fitting data. Did the cycled Si@MnO2@rGO-50°C be fitted with the same equivalent circuit model in Figure 6(a)?
(6) At the same 1 A g-1 rate, the Si@MnO2@rGO-50° electrodes in the rate performance and in the cycling performance give different specific capacity with the range of 100 mAh g-1. What happen?
Round 2
Reviewer 2 Report
The authors have addressed my comments, and have revised the manuscript in detail according to the comments.